# Parental Perspectives on Environmental Factors Affecting Participation of Children with Disabilities: A Scoping Review to Inform Inclusive Healthcare and Support Services

**DOI:** 10.3390/healthcare13111282

**Published:** 2025-05-28

**Authors:** Zeynep Celik Turan, Aleyna Kayim, Anne-Mie Engelen, Kubra Sahadet Sezer, Carolyn Dunford, Esra Aki

**Affiliations:** 1Occupational Therapy Division, Brunel University of London, Uxbridge UB8 3PH, UK; carolyn.dunford@brunel.ac.uk; 2Occupational Therapy Department, Hacettepe University, Ankara 06100, Türkiye; kayimaleyna@gmail.com (A.K.); kubrassezer1@gmail.com (K.S.S.); esraaki@hacettepe.edu.tr (E.A.); 3EQUALITY Research Collective, HOGENT University of Applied Sciences and Arts, 9000 Ghent, Belgium; annemie.engelen@hogent.be

**Keywords:** children with disabilities, parental perspectives, environmental factors, occupational participation, health equity, inclusive services, caregiver experience, pediatric rehabilitation, person–environment–occupation model, scoping review

## Abstract

**Background/Objectives**: Environmental factors influence the participation of children with disabilities in everyday life. Parents, as primary caregivers, provide insights into how these factors support or hinder participation in health-related, educational, personal and social activities. This scoping review aimed to systematically map the literature on parental perceptions of environmental influences on the participation of their children with disabilities. **Methods**: The review followed the PRISMA-ScR guidelines and the five-stage framework by Arksey and O’Malley. Searches were conducted in five electronic databases: MEDLINE, CINAHL Plus, PsycINFO, PsycArticles, and OpenDissertations. Eligible studies were published in English, focused on children aged 0–18 with any type of disability, and reported on parental views of how environmental factors influence occupational participation. Data were charted and analyzed using narrative synthesis and vote-counting. **Results**: Thirty-four studies met the inclusion criteria. Frequently discussed environmental domains included physical, social, and institutional factors, while cultural and economic domains received less attention. Participation was commonly addressed in the contexts of play, education, and social engagement. Most studies used qualitative designs and were conducted in high- and middle-income countries. Standardized tools to assess environmental impacts were rarely employed. **Conclusions**: This review highlights the need for inclusive, family-centered health and social services that address the full range of environmental influences on participation. Future research and policy should prioritize culturally and economically diverse settings, integrate standardized assessment tools, and recognize parental perspectives as essential for designing equitable pediatric healthcare and rehabilitation services.

## 1. Introduction

Children with disabilities are those who, for a variety of reasons, require additional educational support to ensure equal access to learning opportunities and the full development of their potential [1]. Children with disabilities face challenges that affect their participation in daily activities, education, and social interactions [2,3,4]. Participation is influenced by the dynamic interaction of various factors, including the child’s individual characteristics and the environments in which they live, learn, and play [3,5].

Participation refers to involvement in life situations that are meaningful and necessary, such as going to school, playing with peers, or managing personal care. These forms of engagement occur across what are commonly known as occupational performance areas. The Occupational Therapy Practice Framework (OTPF-4) [6] outlines key performance areas including activities of daily living (ADLs), education, play, leisure, social participation, health management, and rest and sleep. These categories provide a useful lens for understanding where and how participation occurs in children’s lives.

The Person–Environment–Occupation (PEO) model emphasizes the interconnectedness of personal, environmental, and occupational domains in shaping occupational performance [7,8]. Within the PEO model, the environment is conceptualized as a multidimensional construct encompassing five contextual domains: the physical environment (e.g., accessibility of spaces, infrastructure); social environment (e.g., interpersonal relationships, social support, attitudes); institutional environment (e.g., policies, availability and flexibility of services); cultural environment (e.g., societal beliefs, values, and norms related to disability); and economic environment (e.g., financial resources, affordability of care and services) [8,9].

Given the complexity and variability of these environmental domains, it is essential to consider how they are experienced and navigated by those closest to the child. Parents, in particular, greatly influence participation at school, at home, and in the community [10]. They navigate their children’s environments and provide valuable insights into how these environments promote or inhibit participation [4,11]. While an accessible physical space may enable participation in community activities, environments perceived by parents as unsafe or stigmatizing may restrict participation [12]. Understanding parental perspectives is critical not only for facilitating participation but also for designing inclusive healthcare services, coordinating support systems, and reducing systemic barriers that hinder child and family well-being [3,11]. However, it is unclear what kind of information is available in the literature about the parents’ perspective on the impact of the environment on the participation of their children with disabilities. For this reason, a scoping review was conducted in order to systematically map the research done in this area. The research question formulated for this review is: what is the parents’ perspective on the impact of the environment on the participation of children with disabilities?

This scoping review aims to explore and synthesize qualitative, quantitative, and mixed-method studies that focus on parents’ perceptions of how environmental factors, as outlined in the PEO model, affect the participation of children with disabilities aged 0–18 years. The findings will inform policies and practices designed to create more inclusive environments, thereby increasing the participation of children with disabilities in daily life, education, and social interactions. This review will map existing evidence to identify key themes, gaps, and opportunities for future research.

## 2. Materials and Methods

### 2.1. Study Design

A scoping review design was preferred over a systematic review as this review did not aim to answer a specific research question; rather, it explores and presents the existing knowledge on the topic of parents’ perspectives on the environment of their children with disabilities [13]. This review was fundamentally guided by the Preferred Reporting Items for Systematic Reviews and Meta-Analyses Extension for Scoping Reviews (PRISMA-ScR) [14] and supplemented by the five-step methodology by Arksey and O’Malley for scoping reviews [15], with the inclusion of Peters et al.’s suggestions [16] considered throughout. This framework consisted of five steps: (i) identifying the research question(s), (ii) identifying relevant studies, (iii) selecting studies, (iv) charting the data, and (v) collating, summarizing, and reporting results.

#### 2.1.1. Framework Stage One: Identifying the Research Question

This scoping review aims to systematically explore and synthesize the available qualitative, quantitative, and mixed-method research that captures parents’ perspectives on how environmental factors, as defined by the PEO model, affect the participation of their children with disabilities, aged 0–18 years, and the extent and state of this evidence. By doing so, it seeks to provide a foundation for future research and inform policy and practice, facilitating the creation of supportive environments.

The research question is: what is the parents’ perspective on the impact of the environment on the participation of children with disabilities and the extent and state of this evidence?

An initial search of MEDLINE, CINAHL, and PsycINFO did not reveal any systematic or scoping reviews that are currently in progress or actively being conducted and are specific to this subject. Then, the Scoping Review Protocol was submitted to the Open Science Framework for registration at the beginning of July 2024.

#### 2.1.2. Framework Stage Two: Identifying Relevant Studies

In collaboration with an academic liaison librarian for the health sciences field, the search strategy was created. A preliminary search of Web of Science, PsychINFO, and MEDLINE was conducted to determine the MeSH index keywords, and 50 articles were produced. Key terms were found during the scoping phase to create an efficient search strategy, as well as from the titles and abstracts of 50 retrieved papers, synonyms of the indices, proposed author keywords, and review team discussions. MEDLINE was used to test a preliminary search approach. After consulting with the review team and the topic librarian, the search technique was improved to ensure the breadth of coverage was maintained and to boost the prevalence of potentially relevant articles. Search phrases included “parent”, “caregiv*”; “perspective”, “thought”; “impact”, “influence”; “home”, “school”, “occupational participation”, “engagement”; “special need”, “disabil*” “impairment”; “children”, “adolescence”, “pediatric”; “rehabilitation”, and “therapy”. The final search strategy was then adjusted for each database based on the various AND/OR combinations of the index terms and keywords. The MEDLINE, CINAHL Plus, PsycINFO, PsycArticles, and OpenDissertations databases were used for the search tactics, which were selected as advised by the Royal College of Occupational Therapists. Please see Appendix A, which outlines the full search strategy for MEDLINE.

#### 2.1.3. Framework Stage Three: Study Selection

Prior to selecting the studies, the review committee engaged in discussions regarding the inclusion criteria. The PCC framework [17] was used to identify these criteria. The review excluded studies that did not satisfy any of these criteria.

PCC Framework:


*Participants*


The main population will include parents of children with disabilities aged zero to eighteen years old. This encompassed a broad range of conditions, including physical disabilities (e.g., cerebral palsy), neurodevelopmental conditions (e.g., autism spectrum disorder, attention-deficit/hyperactivity disorder), intellectual disabilities, sensory impairments (e.g., visual impairment), and chronic health conditions (e.g., epilepsy). Studies were eligible if they focused on this population or included parental perspectives as part of the participant group.


*Concept*


The main concept is the parents’ perspective on the environment’s impact on their children’s participation. Parental perspectives encompass a broad range of views, beliefs, experiences, and attitudes that parents hold regarding various aspects of their children’s lives. Studies including this parent’s perspective on any given aspect of the impacts of the environment and/or environmental factors on the occupational participation of their children will be included in the interview.

OTPF-4 defines participation as “Involvement in a life situation” [6]. In this review, occupational participation will refer to the involvement in meaningful and purposeful activities, often called “occupations”, that are necessary and desired for personal and social fulfillment. These activities can encompass various tasks and roles that individuals engage in daily, such as self-care, education, work, leisure, and social interactions.

In the PEO model [7], occupational domain refers to the groups of tasks a person engages in to meet their self-maintenance, expression, and fulfillment needs. Occupations are “the everyday activities that people do as individuals, in families, and with communities to occupy time and bring meaning and purpose to life. Occupations include things people need to, want to and are expected to do” [18]. Occupations are categorized as basic activities of daily living, instrumental activities of daily living, health management, rest and sleep, education, work, play, leisure, and social participation. Given the focus group of this review is children, education and work, and activities of daily living are considered together.


*Context*


Any setting in which children demonstrate occupational participation will be included. There is no limit on geographic location, gender, race, ethnicity or any other description.

Eligibility Criteria

Studies were included if they met the following criteria: (1) reported on the perspectives of parents of children with disabilities aged 0–18 years; (2) examined how environmental factors influenced the child’s participation in daily life, education, or social activities; (3) employed qualitative, quantitative, or mixed-method designs; and (4) were published in English.

Studies were excluded if they: (1) did not report data specific to parents (e.g., combined child and parent responses without disaggregation); (2) focused solely on interventions, assessments or theory development without examining environmental influences; (3) involved caregiver populations beyond parents (e.g., teachers or therapists) without parental data.

Selection of Sources:

The review search included a broad range of sources, including articles, theses, case reports, reports, conference proceedings, reviews, and clinical trials to consider the diverse perspectives of parents to offer a comprehensive understanding of the environmental factors at play, categorized as physical, cultural, institutional, social, and socio-economic environments, per the PEO model.

The initial search of the databases yielded 5372 sources. Afterward, the duplicates were removed by using the database’s ‘remove duplication’ option, and keywords were screened through the abstract and title. For the purposes of screening and data extraction, 270 recognized citations were compiled and uploaded into a shared OneDrive folder. Two more duplicates were eliminated. Two independent reviewers checked 268 sources’ titles and abstracts against the inclusion and exclusion criteria after a pilot test for benchmarking. Both reviewers had to agree for a study to be included (progressed to the next stage for full text reading) or excluded, and any disputes had to be settled by the third reviewer, who served as chair. Studies that met the study selection criteria were added to the full-text review folder in OneDrive.

After that, 115 studies’ full texts were meticulously examined to assess whether the eligibility criteria were met. Similar to the title and abstract screening, the full texts were also reviewed by two independent reviewers. If there was a disagreement, a third reviewer acted as a chair. While inter-rater agreement was not formally calculated, consistency between reviewers was monitored throughout. During the title and abstract screening stage, 23 articles required adjudication by the third reviewer. At the full-text review stage, 9 articles were referred for final decision. All disagreements were resolved through discussion and consensus. The PRIMSA flow diagram (Figure 1) illustrates the reasons for exclusion of the full-text studies that did not match the inclusion criteria. To make sure all relevant, suitable content was located, a further manual inspection of the reference lists of the included studies was carried out. The last search took place in February 2025. A further 14 studies were retrieved and screened for full text. Five studies were found via citation searching, while 29 were found using primary searches. The review includes 34 papers in total.

#### 2.1.4. Framework Stage Four: Charting the Data

The Microsoft Excel spreadsheet, shared in OneDrive, was used to extract data from the 34 included studies. The review team used an adapted version of the JBI Extraction Template [20] to include important characteristics and the scope of the studies. Similar to the screening phase, one study was charted by all reviewers as a pilot and used for benchmarking when using the template. Author(s), Country, Study Aim(s), Study Design, Participants (N=), Population Characteristics (Appendix B), and Findings were the final headings in the extraction tool. The main reviewer extracted the data from the studies, and a second reviewer confirmed it. If any disputes arose between the main and second reviewers, they were resolved through dialogue with the third reviewer, who acted as a mediator. The charted data were tabulated in Table 1 and Table 2.

#### 2.1.5. Framework Stage Five: Collating, Summarizing, and Reporting the Results

Data were synthesized using the narrative synthesis approach [21], supported by three interrelated strategies: textual description, tabulation, and vote counting. In the initial round of data charting, textual summaries were used to capture detailed findings in line with a structured extraction template (see Appendix B). In a second round of analysis, these summaries were refined into two distinct tables. Table 1 was developed through tabulation to present study characteristics and narrative findings in a concise and consistent format. Table 2 was generated using vote counting to visually represent the presence or absence of each environmental domain and occupational participation area across the included studies. These methods were applied iteratively rather than linearly, with insights from one informing the others. For Table 2, findings were extracted according to PEO’s categorization of environment [9] and the AOTA’s [6] categorization of occupations. In cases where terminology varied in the voting of an occupational participation area or environmental domain looked at within a particular study, they were cross-checked with definitions of AOTA and PEO for validation. A detailed explanation of voting is provided in Appendix C.

**Table 1 healthcare-13-01282-t001:** Description of included studies and narrative synthesis of key findings.

Author, Country (Year)	Study Aims Relevant to This Review	Study Design	Sample Characteristics	Measures Relevant to This Review	Key Findings
Agnew et al., Australia (2024) [22]	To explore parents’ experiences of how assistance dogs influence their children’s occupational participation and engagement with autism.	Exploratory qualitative design	- 6 mothers - 7 children with autism, aged 4 to 11 years	Semi-structured interviews	Parents reported that assistance dogs increased their children’s participation in social, community, and leisure activities. The dogs supported confidence in public, enabled more outings, and encouraged peer interaction. Parents also described enhanced emotional regulation and reduced anxiety, which facilitated greater engagement in play, school, and daily routines.
Alavi et al., UK (2012) [23]	The study’s primary aim was to develop a conceptual model representing the impact of musculoskeletal impairments (MSIs) in the lives of children in Malawi, based on empirical data from children, their families, and community stakeholders.	Descriptive qualitative design	- 56 parents - 34 children with MSIs, aged 2 to 10 years	Semi-structured interviews	Parents reported that environmental barriers, including inaccessible infrastructure, negative social attitudes, and poverty, restricted their children’s participation in school, play, and household activities. They highlighted exclusion by peers and teachers, physical pain, and being left alone or behind as common consequences of environmental challenges faced by their children.
Anaby et al., Switzerland (2017) [24]	This study aimed to explore parents’ perspectives on the PREP (Pathways and Resources for Engagement and Participation) intervention, which focuses on removing environmental barriers to support youth with physical disabilities (PD) in participating in chosen community leisure activities. The goal was to understand the intervention’s perceived impact and process.	Descriptive qualitative design	- 12 parents (20 mothers, two fathers) - 12 youth with PD, aged 12 to 18 years	Semi-structured interviews	Parents reported that removing environmental barriers, such as inaccessible spaces, financial constraints, and unsupportive attitudes, enabled their children to participate more fully in leisure and community activities. They observed improvements in physical abilities, emotional well-being, social interaction, and autonomy, highlighting the value of individualized, environment-focused support provided through the PREP intervention.
Bevans et al., USA (2020) [25]	The study aimed to evaluate the Participation and Sensory Environment Questionnaire–Home Scale (PSEQ–H) psychometric properties—a parent-report tool measuring how the sensory environment affects young children’s participation in home-based activities.	Psychometric validation study design	- 304 parents - 305 children (167 with Autism Spectrum Disorder (ASD), 137 neurotypical), aged 2 to 7 years	PSEQ–H	Parents reported that the sensory environment impacted their children’s participation in dressing, self-care, play, and sleep at home. Children with ASD experienced greater participation challenges than their neurotypical peers.
Biyik et al., Turkey (2021) [26]	Using the ICF-CY framework and a structured parent-report questionnaire, this study examined parents’ perspectives on how the COVID-19 stay-at-home period affected the body functions, activity and participation levels, and environmental factors related to children with cerebral palsy (CP).	Descriptive survey design	- 103 parents (89 mothers, 14 fathers) - 103 children with CP, aged 2 to 18 years	Custom-developed parental questionnaire	Parents reported reduced participation in daily activities during the COVID-19 lockdown, especially in mobility, self-care, social play, and outdoor engagement. Limited access to rehabilitation and professional support shifted responsibility to families, which overwhelmed caregivers and further restricted children’s opportunities to engage in meaningful, developmentally appropriate activities at home.
Brooke Willis, USA (2016) [27]	The study aims to identify social opportunities for children with disabilities in the community. It seeks to enhance parents’ understanding of the benefits of social participation for their children. The research reviews barriers and facilitators affecting social participation among children with disabilities.	Pilot intervention study with a pre-post descriptive design	- 4 mothers - 4 school-aged children with disabilities	Participation and Environment Measure for Children and Youth (PEM-CY)	Parents perceived that environmental barriers significantly limited their children’s social participation, particularly due to a lack of inclusive programs, limited community resources, and negative social attitudes. They also reported feeling underprepared and unsupported, but after the intervention, they expressed increased confidence, awareness, and ability to access resources and support their child’s participation more effectively.
Egilson Snaefridur et al., Iceland (2016) [28]	The study aimed to explore parent perspectives on participation patterns and environmental supports and barriers for high-functioning children with ASD compared to children without ASD, regarding their child’s participation in leisure activities, in the context of a structured movement program.	Cross-sectional quantitative design	- 300 parents (99 of children with ASD, 241 of without ASD) - 300 children, aged 8 to 17 years	PEM-CY	Parents of children with ASD perceived lower community participation and fewer environmental supports compared to peers. They identified more barriers and inadequate resources, linking these to reduced satisfaction. These findings underscore the critical role of environmental features in shaping children’s opportunities for meaningful community participation.
Egilson Snaefridur et al., Iceland (2018) [29]	To explore and compare parent perspectives on the participation of high-functioning children with and without ASD in home activities, environmental features affecting participation, and strategies parents use to support participation at home.	Mixed methods design	- 300 parents (99 of children with ASD, 241 of without ASD) - 300 children, aged 8 to 17 years	PEM-CY	Parents of children with ASD report that home environments can support or limit participation, especially in social and daily activities. Limited resources and support reduce engagement. Parents use tailored strategies to help, but they strongly need more supportive environments and accessible resources.
Eicher et al., USA (2017) [30]	The study explored parents’ experiences and expectations of raising a child with sensorimotor impairments regarding their child’s participation in leisure activities, in the context of a structured movement program.	Phenomenological qualitative design	- 6 parent dyads - 6 children with sensorimotor impairments, aged between 5 and 8 years	Semi-structured interviews	Supportive relationships, school staff attitudes, cultural norms, service access, and assistive technology all impact participation. Factors like teacher–student fit, program quality, parental expectations, and system flexibility shape how well children with sensory needs engage in activities.
Fuentes et al., USA (2019) [31]	The study aimed to explore how traditional culture influences health, disability, and healthcare services among American Indian and Alaska Native (AI/AN) children and youth with disabilities, and to focus on participation in cultural activities and family experiences with service systems.	Exploratory descriptive qualitative design	- 13 parents (8 mothers, five fathers) - 13 children with disabilities, aged 6 to 17 years	Semi-structured interviews	Cultural participation is vital for well-being, but functional challenges and environmental barriers often hinder access. Healthcare providers frequently overlook culturally specific needs, and families may hesitate to share them, as such needs are rarely recognized as part of rehabilitation.
Galvin et al., Australia (2010) [32]	This study aimed to examine parents’ perceptions of their children’s participation in home, school, and community settings following acquired brain injury (ABI) and to explore the impact of environmental and child-related factors on participation levels.	Descriptive cross-sectional design	- 20 parents - 20 children with ABI, aged 64 to 184 months	- Child and Family Follow-up Survey (CFFS), including - Child and Adolescent Scale of Environment (CASE)	Parents identified environmental barriers, such as limited school support, peer exclusion, and reduced access to rehabilitation, as key factors restricting their children’s participation in school, social, and recreational activities. They also described their active role in supporting engagement and emphasized the need for tailored, responsive services post-brain injury.
Ghaffari et al., Iran (2020) [33]	To test a theoretical model including child, family, and environmental variables to identify predictors of the intensity of leisure participation among children with CP in Iran.	Cross-sectional design	- 232 parent - 232 children with CP, aged 6 to 14	- Children’s Assessment of Participation and Enjoyment (CAPE) - Craig Hospital Inventory of Environmental Factors (CHIEF) - Family Environment Scale (FES)	Parents’ knowledge and interest in recreation supported their children’s participation. Family dynamics such as cohesion, organization, and conflict influenced activity patterns. Environmental barriers, including attitudes and lack of support, were seen as key factors limiting children’s engagement in leisure activities.
Gothwal et al., India (2022) [34]	The study aimed to explore how the COVID-19 lockdown impacted the lives and educational participation of visually impaired (VI) school-age children in India, from the perspective of parents, particularly regarding access to online learning, support, and environmental barriers.	Descriptive qualitative design	- 48 parents - 48 children with VI, aged 7 to 19 years	Custom-developed open-ended survey	Parents reported that the COVID-19 lockdown severely disrupted their children’s education, social interaction, and daily routines. Limited access to online learning, lack of assistive technology, and absence of teacher support restricted participation. Emotional stress, isolation, and uncertainty further impacted children’s engagement in both academic and social activities.
Handberg et al., Denmark (2021) [35]	This study aimed to explore parents’ perspectives on how the COVID-19 pandemic affected the health, daily participation, and quality of life of children with neuromuscular diseases (NMD) in Denmark, with particular attention to changes in rehabilitation, education, social activities, and environmental supports.	Descriptive cross-sectional survey	- 67 parents (54 mothers, 13 fathers) - 67 children with NMD, aged 3 to 15 years	Custom-developed parental questionnaire	Parents reported that children with NMD faced reduced participation in education, leisure, and social activities due to pandemic-related restrictions. Disrupted rehabilitation, increased home isolation, and fear of infection limited engagement. Parents’ heightened risk perception was linked with greater child isolation, anxiety, and reduced opportunities for meaningful interaction and activity.
Heah et. al, Canada (2006) [36]	To explore what successful participation in non-school activities means to children with PD and neurological disabilities and their parents, and to identify the personal and environmental supports and barriers influencing their participation.	Descriptive phenomenological qualitative study	- 8 parents (7 mothers, one father) - 8 children with disabilities, aged 6 to 15.6 years	Semi-structured interviews	Parents viewed successful participation as engagement in personally meaningful activities, often involving social connection, independence, and enjoyment. Environmental facilitators included supportive relationships, inclusive community programs, and parental advocacy. Barriers included inaccessible environments, societal stigma, and a lack of suitable opportunities. Parent values and expectations strongly influenced participation choices.
Hong et al., UK (2022) [37]	This study examined differences in home participation patterns and environmental factors between Korean children with and without disabilities. It explored how often children participate in home-based activities, how involved they are, and which environmental factors parents perceive as barriers or supports to their child’s participation.	Quantitative cross-sectional design	- 80 parents of children with disabilities (74 mothers, six fathers) and 104 parents of children without disabilities - 80 children with disabilities and 104 children without disabilities, aged 5 to 13 years	The Korean PEM-CY	Parents of children with disabilities reported lower frequency and involvement in most home activities and expressed a stronger desire for change. Environmental barriers included activities’ physical, cognitive, and social demands, inadequate supplies, and limited financial resources. These factors contributed to reduced participation, especially in school-related and social activities.
Jaarsma et al., Netherlands (2015) [38]	Parents identified the greatest needs in school participation, leisure activities, and access to assistive devices. Many needs were unmet, and those related to advocacy, rights, and tailored information were rated as high priorities. Child functioning and parental mental health were associated with the number and types of expressed needs.	Mixed methods design	- 38 parents (most were mothers) - 30 children with PD, aged 8 to 20 years old	PEM-CY	Parents reported that supportive peers, enjoyment, and accessible programs facilitated sports participation, while barriers included inaccessible locations, lack of information, equipment needs, and limited inclusive options. Professionals highlighted the need for better service coordination. Children valued fun and friendship, but physical fatigue and dependency on others often limited their engagement.
Kang et al., Taiwan (2017) [39]	The study aimed to identify the environmental barriers perceived by parents of preschool children with and without PD in Taiwan and to compare the impact of these barriers on children’s participation across home, preschool, and community settings.	Cross-sectional comparative design	- 142 parents of children with PD - 192 parents of preschool children with typical development (TD) - Children were aged 2 to 6 years	CASE-C	Parents of preschool children with PD reported more environmental barriers than those of TD children. These included limited access to services, insufficient resources, and negative social attitudes. Such barriers affected children’s participation and inclusion in daily activities across home, school, and community settings.
Khetani et al., Canada (2018) [40]	The study’s primary aim was to explore changes in home participation frequency and involvement among critically ill children during the first 6 months after discharge from the pediatric intensive care unit (PICU). The secondary aim was to identify child, service, and environmental factors that predict changes in home participation frequency and involvement in the home setting.	Prospective longitudinal cohort study design	- 180 parents - 180 children with various diagnoses who were admitted to the PICU, aged 1 to 17 years	Participation and Environment Measure (PEM)	Caregivers observed that children with more severe conditions before PICU admission improved more in home participation, with home support and environmental modifications aiding recovery. Rehabilitation interventions targeting functional capabilities and home environment may be viable approaches during the early recovery phase. Environmental interventions may be more time-efficient after a PICU stay.
Law et al., Canada (2007) [41]	The study aims to describe parents’ perceptions of environmental barriers to participation for children with PD. It focuses on these children’s recreational, community, and school participation challenges.	Cross-sectional quantitative design	- 427 parents (54 male, 46 female) - 427 children with PD, aged between 6 and 14 years	CHIEF	Parents reported significant psychosocial barriers to their children’s participation, often linked to emotional and behavioral challenges. Past discrimination and negative community experiences reduced children’s willingness to engage, especially in social settings. Physical barriers were less commonly noted compared to psychosocial factors.
Lawlor et al., UK (2006) [42]	To identify features of the physical, social, and attitudinal environments that facilitate or restrict participation for children with CP, as reported by families.	Descriptive qualitative design	- 13 families (5 mothers, three fathers, three both parents in the scope of this review) - 13 children with CP, aged 5 to 17 years old	Semi-structured interviews	Parents identified physical access and bureaucratic delays as key barriers to their children’s participation, but rarely noted facilitators like improved resources. This suggests a tendency to accept environmental limitations rather than advocate for change, reflecting concerns about systemic and corporate obstacles to inclusion.
Maddocks et al., South Africa (2020) [43]	This study explored caregivers’ perceptions of raising children with HIV-related disabilities in a resource-poor South African community, focusing on caregiving challenges, access to rehabilitation, and environmental factors affecting children’s participation and well-being.	Interpretive qualitative design	- 8 parents (7 mothers, one father) - 10 children living with HIV, aged 6–10 years	Semi-structured interviews	Parents described major environmental barriers affecting their children’s participation, including inaccessible housing and transport, exclusion from school, and lack of rehabilitation services. They expressed emotional distress, fear for their children’s futures, and frustration over institutional failures. Social stigma around HIV and disability further isolated both the parent and child.
Manitsa et al., UK (2024) [44]	This study explored parents’ perspectives on how habilitation services support the participation, education, and socio-emotional development of children and adolescents with VI, particularly in promoting independence, accessibility, and inclusion in daily and school life.	Interpretive qualitative design	- 16 parents (12 female and four male) - 16 children with VI	Semi-structured interviews	Parents reported that habilitation services enhanced their children’s participation by promoting independence, confidence, and mobility. They valued the role of habilitation workers in supporting education, community access, and social interaction. However, service inconsistencies and limited availability across regions created unequal participation and skill development opportunities.
Marcone et al., Italy (2023) [45]	To explore changes in parental perceptions of support and participation opportunities for children with ASD and intellectual disability (ID) during the COVID-19 lockdown in Italy, with attention to rehabilitation, schooling, and social services.	Descriptive cross-sectional design	- 106 parents (92 mothers and 14 fathers) - 106 children with ID and/or ASD	-Custom-developed open-ended survey	Parents reported that the COVID-19 lockdown severely limited their children’s education, therapy, and social life participation. Reduced institutional support and peer interaction led to fewer opportunities for meaningful engagement in daily routines and community activities, increasing parental stress and the burden of facilitating participation at home.
Mei et al., UK (2015) [46]	This study explored parents’ perspectives on the activities and participation of their children with CP, aged 4–10 years, across home, school, and community settings. Using the ICF-CY framework, it sought to understand environmental and personal factors influencing children’s everyday participation	Descriptive qualitative study	- 13 parents (11 mothers and two fathers) - 13 children with CP, aged 4 to 10 years	Semi-structured interviews	Parents described various environmental and personal factors influencing their children’s participation, including communication challenges, inaccessible environments, and negative social attitudes. Facilitators included supportive peers, familiar routines, and parental involvement. Communication was central to independence, social interaction, and meaningful engagement across home, school, and community settings.
Nithya et al., India (2021) [47]	The study aimed to systematically assess the impact of COVID-19 on activities of daily living in children with autism. It also focused on evaluating changes in play behaviors of children with ASD during the pandemic.	Cross-sectional survey design	- 100 parents (88 female, 12 male) - 100 children with ASD, aged 2 to 16 years	Custom-developed parental survey	During COVID-19, parents of children with ASD reported disrupted routines, irregular sleep, increased screen time, reduced interactive play, greater social withdrawal, and lower physical activity. These changes significantly impacted daily behaviors and participation.
Njelesani et al., Canada (2015) [48]	The study aimed to explore barriers perceived by parents of children with developmental disabilities to their children’s engagement in physical activity. It sought to understand how environmental, personal, and contextual factors shaped children’s opportunities for active engagement.	Descriptive qualitative design	- 9 parents (5 mothers, 2 fathers, and 1 parent pair) - 9 children with developmental disabilities, aged 10 to 17 years	Semi-structured interviews	Parents identified time constraints, inaccessible environments, limited programs, and financial barriers as key obstacles to their children’s participation in physical activity. They also emphasized the need for individualized activities suited to their child’s abilities and noted that social discomfort and safety concerns further limited meaningful engagement in physical activities.
Piskur et al., Netherlands 2014) [10]	To provide an overview of the number, domains, and priorities of needs expressed by parents to support the participation of their school-aged child with a PD. The study also investigated how these needs relate to child and family characteristics.	Cross-sectional survey design	- 146 parents (84.9% mothers) - 146 children with disabilities, aged 4 to 12 years	- Family Needs Inventory—Pediatric Rehabilitation - Family Report Questionnaire	Parents identified the greatest needs in school participation, leisure activities, and access to assistive devices. Many needs were unmet, and those related to advocacy, rights, and tailored information were rated as high priorities. Child functioning and parental mental health were associated with the number and type of expressed needs.
Rosenberg et al., Canada (2011) [49]	The study aimed to assess parents’ perceptions of environmental factors as barriers to their child’s participation in activities. The research aimed to support the inclusion of environmental restrictions in child evaluation processes for effective intervention programs.	Descriptive quantitative design	- 78 parents - 78 children with mild developmental disabilities, mean age 5.20 ± 0.52 years	Environmental Restriction Questionnaire (ERQ)	Parents saw home human factors as more limiting than physical ones, though overall, the home was viewed as less restrictive than school or community settings. Key barriers included family income and craft space at home, partner’s occupation at school, and neighborhood safety and traffic in the community.
Shields et al., Australia (2022) [50]	The study aimed to investigate modifiable child and caregiver factors influencing community participation among children with Down syndrome.	Cross-sectional quantitative design	- 89 parents (54 females, 35 males) - 89 children with Down syndrome, aged 5 to 18 years	PEM-CY	Parents saw themselves as barriers to community participation due to limited time and daily responsibilities. Caregiver availability was linked to more frequent attendance in activities. Factors like the child’s functional ability, health, and behavior also influenced their friendships and hobbies.
Shuttleworth et al., Australia (2024) [51]	To explore parents’ experiences and perceptions of the barriers and facilitators affecting their child’s participation in gymnastics, to inform more inclusive environments and pathways to engagement in gymnastics for children with disabilities.	Sequential explanatory mixed-methods design	- 58 parents - 45 children with disabilities, mean age 10 ± 4.6 years	- Custom-developed parental survey - Semi-structured interviews	Parents identified that inclusive environments, knowledgeable coaches, and enjoyment were key to sustaining gymnastics participation. Sensory overload, coach turnover, and lack of funding were barriers. Gymnastics supported children’s physical and social development, particularly for those with autism. Parents valued individualized approaches over competition and sought greater visibility of inclusive options.
Towns et al., England (2022) [52]	The study’s primary aim is to explore how balance confidence and emotional responses to balance loss affect physical activity participation among youth with CP across different GMFCS levels.	Descriptive qualitative study design	- 8 parents (5 mothers, three fathers) - 8 youth with CP, aged 9 to 17 years	Semi-structured interviews	Parents noted that social support, balance confidence, and access to adaptive equipment were key factors influencing physical activity participation, with youth in higher GMFCS levels showing more reluctance due to peer concerns.
Varengue et al., France (2022) [53]	To explore parents’ perceptions of how the COVID-19 lockdown affected the daily activities and well-being (morale, behavior, social interaction, schooling, and physical activity) of children with PD in France, and to compare these with those of TD children.	Cross-sectional survey design	- 1376 parents(86.8% mothers, 12.3% fathers) - 1367 children with PD, aged 1 to 18 years	Enfant Confinement Handicap BesOins (ECHO) survey	Parents reported that the COVID-19 lockdown significantly disrupted their children’s participation in physical activity, schooling, and social interaction. Loss of rehabilitation services, limited support, and inaccessible environments increased caregiving demands. Families of children with PD experienced more negative impacts than families of TD children across all participation domains.
Warnink-Kavelaars et al., Netherlands (2019) [54]	Using the ICF-CY framework to explore parents’ perspectives on how Marfan syndrome (MFS) affects the daily functioning of their children, as well as the broader impacts on parental and family life.	Descriptive qualitative study design	- 26 parents (10 in interviews and 16 in focus groups) - 24 children with MFS (8 in interviews, 16 in focus groups), aged 4–12 years	- Semi-structured interviews - Focus groups	Parents reported restricted participation in school, sports, play, and leisure, leading to feelings of difference and encountering unsupportive attitudes. Key ICF-CY environmental factors included support from family and teachers (e3), societal attitudes (e4), access to services (e5), and assistive products like shoes, splints, and wheelchairs (e1).

**Table 2 healthcare-13-01282-t002:** Vote counting: environmental domains and occupational participation areas in the included studies.

	Environments						Occupational Participation Areas					
Author, Country (Year)	Physical	Cultural	Social	Institutional	Economic	ADLs	Health Management	Rest and Sleep	Education/Work	Play	Leisure	Social Participation
Agnew et al., Australia (2024) [22]	x	-	x	-	-	x	-	x	x	x	x	x
Alavi et al., UK (2012) [23]	x	-	x	x	x	x	x	x	x	x	x	x
Anaby et al., Switzerland (2017) [24]	x	-	x	x	x	x	-	-	-	-	x	x
Bevans et al., USA (2020) [25]	x	-	x	-	-	x	-	x	-	x	-	x
Biyik et al., Turkey (2021) [26]	x	-	x	x	-	x	x	x	-	x	-	x
Brooke Willis, USA (2016) [27]	x	-	x	x	-	-	-	-	x	x	x	x
Egilson Snaefridur et al., Iceland (2016) [28]	x	-	x	x	x	x	x	-	-	x	x	x
Egilson Snaefridur et al., Iceland (2018) [29]	x	-	x	x	x	x	x	-	x	x	x	x
Eicher et al., USA (2017) [30]	x	x	x	x	-	x	x	-	x	x	x	x
Fuentes et al., USA (2019) [31]	x	x	x	x	-	x	x	x	x	x	x	x
Galvin et al., Australia (2010) [32]	x	-	x	x	-	x	x	-	x	x	x	x
Ghaffari et al., Iran (2020) [33]	x	-	x	x	x	-	x	-	-	x	x	x
Gothwal et al., India (2022) [34]	x	-	x	x	-	-	-	-	x	-	x	x
Handberg et al., Denmark (2021) [35]	x	-	x	x	-	-	x	-	x	-	x	x
Heah et. al, Canada (2006) [36]	x	x	x	x	-	x	x	x	-	x	x	x
Hong et al., UK (2022) [37]	x	-	x	-	x	x	-	-	x	x	x	x
Jaarsma et al., Netherlands (2015) [38]	x	x	x	x	x	-	x	-	x	x	x	x
Kang et al., Taiwan (2017) [39]	x	-	x	x	-	x	-	-	x	-	-	x
Khetani et al., Canada (2018) [40]	x	-	x	x	-	x	x	-	-	x	x	x
Law et al., Canada (2007) [41]	x	-	x	x	x	x	x	-	x	x	x	x
Lawlor et al., UK (2006) [42]	x	-	x	x	x	x	x	-	x	x	x	x
Maddocks et al., South Africa (2020) [43]	x	-	x	x	x	x	x	-	x	x	-	x
Manitsa et al., UK (2024) [44]	x	-	x	x	-	x	-	-	x	-	x	x
Marcone et al., Italy (2023) [45]	x	-	x	x	-	-	x	-	x	-	x	x
Mei et al., UK (2015) [46]	x	-	x	x	-	x	x	-	x	x	x	x
Nithya et al., India (2021) [47]	x	-	x	x	-	x	x	x	x	x	-	x
Njelesani et al., Canada (2015) [48]	x	-	x	x	x	-	-	-	x	x	x	x
Piskur et al., Netherlands 2014) [10]	x	-	x	x	x	x	x	-	x	x	x	x
Rosenberg et al., Canada (2011) [49]	x	x	x	x	x	-	-	-	x	x	x	x
Shields et al., Australia (2022) [50]	x	-	x	x	x	-	x	x	x	x	x	x
Shuttleworth et al., Australia (2024) [51]	x	x	x	x	x	-	x	-	-	x	x	x
Towns et al., England (2022) [52]	x	-	x	-	-	x	x	-	x	x	x	x
Varengue et al., France (2022) [53]	x	-	x	x	-	x	x	-	x	-	x	x
Warnink-Kavelaars et al., Netherlands (2019) [54]	x	-	x	x	x	x	x	-	x	x	x	x
**Total**	**34**	**6**	**34**	**30**	**16**	**24**	**24**	**8**	**26**	**26**	**29**	**34**

x means the domain was a part of the named study, - means the domain was not a part of the named study. Bold section indicates the total number of this domain explored in the studies included in this scoping review.

## 3. Results

Data from the included studies were systematically extracted using a standardized template, and key findings were synthesized through a combination of narrative synthesis (Table 1) and vote-counting (Table 2). This dual approach allowed both the depth and breadth of the data to be represented, capturing not only the diversity of perspectives but also the frequency with which different environmental and occupational domains were addressed.

The synthesis revealed that the most commonly examined environmental domains were physical (in all 34 studies), followed by social and institutional domains. In contrast, cultural and economic environments were explored less frequently. With regard to occupational participation, the most often discussed areas were social participation, education, and play and leisure, while fewer studies addressed ADLs, health management, and rest and sleep. The findings highlighted a broad consensus that environmental features, ranging from physical accessibility to social attitudes and policy structures, play a significant role in shaping the opportunities and challenges experienced by children with disabilities. These patterns provided the foundation for the thematic structuring of the results that follow.

### 3.1. Description of Included Studies

This scoping review included 34 studies published across a range of international journals, exploring the perspectives of parents regarding how environmental factors influence the participation of children with disabilities. The studies spanned a variety of geographical contexts, including Europe, North America, Asia, Africa, and Australia, reflecting a diverse range of socio-cultural and institutional environments. The publication years ranged from the 2000s to 2023, with the majority of studies conducted in the past decade, indicating a growing research interest in the intersection between environment, disability, and participation.

Most studies were conducted in high- or middle-income countries, such as the United States, Canada, Australia, the United Kingdom, South Korea, and several European countries. However, a few studies also provided valuable insights from low-resource settings such as Malawi, Trinidad and Tobago, and South Africa, highlighting global differences in environmental barriers and supports and exploring the relationship between environmental factors and occupational participation, focusing on the perspectives and experiences of parents or primary caregivers.

### 3.2. Participant Characteristics

Participants in the reviewed studies included parents of children with disabilities. Sample sizes varied widely, ranging from small, in-depth studies with fewer than 20 participants to large-scale surveys involving over 100 families. In some cases, data were collected from parents and children to capture multi-perspective insights; however, only data reflecting parental perspectives were extracted and analyzed in this review. Children’s ages span from toddlers to adolescents, reflecting the broad developmental spectrum in the literature. In studies where gender was specified, mothers represented the majority of respondents, often comprising over 80% of the sample, reflecting traditional caregiving roles in many cultural contexts.

The children had a variety of diagnoses, such as ASD, CP, Down syndrome, intellectual disabilities, and other physical or developmental conditions. The age of children ranged from infancy to adolescence (approximately 2 to 18 years). Many studies focused on specific age brackets, such as preschool-aged children (2–5 years), school-aged children (6–12 years), or adolescents (13–18 years), depending on the participation domains of interest. Several studies included mixed diagnostic groups, while others focused on specific conditions. Some studies also compared participation between children with disabilities and their typically developing peers or siblings. Some studies included a broader range to capture developmental transitions and their implications for participation.

### 3.3. Environmental Domains

The studies included in this review explored environmental components using five domains of the PEO framework: physical, social, institutional, cultural, and economic. While each study emphasized different combinations of these domains, many highlighted their interconnection in shaping opportunities for meaningful participation.

The relationship between environmental domains and occupational participation areas was a central focus across the included studies. Of the five domains, the physical and social environment was addressed in all 34 studies, followed by the institutional environment addressed in 30. In contrast, the cultural and economic domains were represented in 6 and 16 studies, respectively.

#### 3.3.1. Physical Environment

The physical environment was the most consistently addressed domain across studies, discussed in all 34 studies. Parents described how inaccessible housing, school buildings, and public spaces hindered their children’s ability to engage in daily activities. Specific challenges included a lack of lifts, adapted bathrooms, uneven terrain, and the absence of inclusive play spaces. Sensory features of the physical environment, such as lighting, sound, and crowdedness, were particularly noted in studies involving children with autism spectrum disorder and sensory processing differences.

#### 3.3.2. Social Environment

The social environment was another domain reported in all 34 studies. Supportive relationships with teachers, peers, and community members were perceived as facilitators, while social exclusion, bullying, or negative societal attitudes were seen as barriers. Studies highlighted that participation was shaped not only by what children could access physically but also by how they were perceived and included socially. Family dynamics and broader community interactions played a significant role in either promoting or limiting engagement.

#### 3.3.3. Institutional Environment

The institutional environment was discussed in 30 studies in relation to the structure, availability, and flexibility of services such as health, education, and social care. Barriers reported by parents included long waiting times, eligibility criteria, lack of coordination between services, and inadequate accommodations in educational settings. Some parents also described taking on advocacy roles or navigating bureaucratic systems to ensure support for their children. Institutional responses to crises, such as the COVID-19 pandemic, were also noted as having significant effects on service continuity and access.

#### 3.3.4. Cultural Environment

The cultural environment, though only explored in six studies, highlighted how societal norms, values, and beliefs influenced children’s participation. In some contexts, disability was associated with stigma or viewed as a private matter, reducing opportunities for public engagement. In others, cultural expectations shaped parenting roles or limited the types of activities deemed appropriate for children with disabilities.

#### 3.3.5. Economic Environment

The economic environment was addressed in 16 studies that considered the role of financial resources in enabling or restricting participation. Parents reported that limited household income, the cost of assistive devices or private services, and lack of transportation options created barriers to participation. Economic disparities were also linked to inequitable access to educational and healthcare services.

Together, these studies illustrate that the environments in which children live, learn, and play are not neutral backdrops. Instead, they are dynamic and often inequitable systems that either enable or constrain participation depending on how well they align with each child’s abilities, needs, and aspirations.

### 3.4. Occupational Participation Areas

Across the 34 studies included in this review, various occupational participation areas were explored in relation to environmental influences. According to the synthesis in Table 2, the most frequently addressed domains were social participation (34 studies), education/work (26 studies), play (26 studies), and leisure (29 studies). Activities of daily living (ADLs) were considered in 24 studies, followed by health management (24 studies) and rest and sleep (8 studies). The following section summarizes how parents perceived environmental influences across each occupational domain.

#### 3.4.1. Activities of Daily Living (ADLs)

Parents reported that environmental barriers affected their children’s ability to perform daily self-care tasks such as dressing, toileting, and feeding. Physical constraints like narrow doorways, inaccessible bathrooms, or the absence of assistive devices were frequently mentioned. In several cases, institutional factors such as the availability of home-based services or occupational therapy input also influenced children’s independence in ADLs.

#### 3.4.2. Education/Work

Environmental influences on education were prominent. Parents discussed school accessibility, availability of specialist support, teacher attitudes, and inclusive education policies. Positive school environments were seen to enhance participation, whereas physical inaccessibility, rigid curricula, or negative attitudes created barriers.

#### 3.4.3. Play and Leisure

Parents identified limited access to inclusive play spaces, lack of adapted recreational programs, and sensory overstimulation as environmental constraints. Both physical and social environments were seen as shaping opportunities for participation in enjoyable and developmentally meaningful activities.

#### 3.4.4. Health Management

Seven studies addressed health management routines. Parents highlighted transportation issues, inconsistent scheduling, financial constraints, and poor coordination between services as environmental challenges that disrupted therapy or medical care routines. Institutional flexibility and support were seen as facilitators for maintaining health routines.

#### 3.4.5. Social Participation

Social participation was the most frequently addressed area. Parents described the role of peer acceptance, inclusive social attitudes, and community programs as supportive of engagement, while social stigma, bullying, or exclusion from group activities served as barriers. Cultural norms were also noted as shaping social opportunities, particularly in public spaces.

#### 3.4.6. Rest and Sleep

Although mentioned in fewer studies, rest and sleep were recognized as being affected by household layouts, shared bedrooms, noise levels, or lack of sensory regulation. These factors were often discussed in relation to the broader home environment and family routines.

## 4. Discussion

This scoping review examined how parents perceive the influence of environmental factors on the occupational participation of children with disabilities. Anchored in the Person–Environment–Occupation (PEO) model, the review synthesized qualitative, quantitative, and mixed-method studies that explored five key environmental domains: physical, social, institutional, cultural, and economic. While the reviewed literature was diverse in geographic scope and methodological approach, the synthesis reveals important patterns in how parents experience, describe, and navigate environmental facilitators and barriers. This section interprets the findings by rigorously examining the visibility and distribution of environmental domains, the mismatch between systemic conditions and children’s needs, the methodological limitations shaping current evidence, and the positive environmental features that parents perceive as enabling participation.

A dominant pattern across the studies was the uneven attention given to different environmental domains. The physical and social environments were discussed in all included studies, indicating that these domains are more easily recognized by parents and more frequently investigated by researchers (e.g., [22,28]). Physical accessibility, infrastructure, and sensory components were often described as shaping participation in everyday occupations by either promoting or restricting participation. Likewise, relationships with teachers, peers, and extended family members were repeatedly cited as enabling or hindering children’s engagement in education, play, and social routines.

In contrast, cultural and economic environments were significantly underrepresented. Where discussed, these domains revealed complex challenges, such as stigma, social expectations, limited financial resources, and access disparities (e.g., [31,43]). The relative absence of cultural and economic domains raises questions about the assumptions guiding participation research. Domains that are harder to observe or measure may be sidelined in favor of those more easily quantified or addressed through environmental modification. This discrepancy suggests the need for a more holistic and inclusive approach that values the full spectrum of environmental influences as experienced and articulated by parents.

In multiple studies, parents identified various systemic conditions, including inflexible or rigid institutional policies, under-resourced services, and insufficient intersectoral coordination, that have restricted their children’s occupational participation (e.g., [32,45]). Parents’ narratives revealed unmet needs in institutional support and inequitable access to services, suggesting that healthcare systems must shift toward more coordinated, culturally competent, and economically sensitive models of care. Despite the systemic nature of these challenges, few studies explored how institutional barriers might be mitigated through policy-level reforms or integrated service models. This disconnection between the scope of environmental barriers and the focus of interventions was an ongoing concern across the literature.

Numerous studies highlighted how families undertake additional responsibilities to establish or sustain environments that support their children’s participation. These included navigating bureaucratic systems, modifying the home setting, or proactively engaging with educators and service providers to secure appropriate accommodations (e.g., [36,42]). Despite being systemically underrepresented in formal decision-making, parents act as de facto care coordinators, navigating fragmented services and advocating for their children’s needs, underscoring the need to treat caregiver experience as an essential data source in health service design. These findings illustrate how families respond to environmental inadequacies in ways that directly affect their children’s participation in daily occupations, particularly in education, health management, and social inclusion.

In the reviewed literature, the methodological choices observed reflect broader strains of how environmental factors are conceptualized and assessed. Most studies employed qualitative designs, particularly semi-structured interviews, which were often informed by the International Classification of Functioning, Disability and Health (ICF) [55] framework. These approaches allowed for in-depth insights into parents’ lived experiences but varied widely in how comprehensively they explored all five environmental domains. While the ICF served as a guiding framework in many of the included studies, particularly those using qualitative methodologies, the present review adopted the PEO model to structure the synthesis. The ICF offers a globally standardized language for describing functioning and disability, but its environmental classification is broad and often oriented toward system-level comparison. In contrast, the PEO model emphasizes the dynamic interaction between individuals and their environments, with a central focus on occupation and participation. This model allowed for a more context-sensitive and client-informed analysis of how parents described facilitators and barriers to participation. Its categorization of the environment into five distinct domains enabled a nuanced yet systematic mapping of parental perspectives across diverse settings. In this way, the PEO framework supported both conceptual clarity and practical relevance in interpreting the environmental dimensions of participation as described in the reviewed studies.

Only a minority of studies used standardized instruments. Among these, the Participation and Environment Measure for Children and Youth (PEM-CY) [56] was the most frequently used, appearing in six studies. Other instruments such as the Child and Adolescent Scale of Environment (CASE) [57], Children’s Assessment of Participation and Enjoyment (CAPE) [58], and Child and Family Follow-up Survey (CHIEF) [59] were used infrequently. While these tools offer structured insights, they also have notable limitations. For example, the PEM-CY focuses heavily on physical and social environments, providing limited attention to cultural and economic contexts. The CHIEF captures environmental barriers but does not address the relational or functional aspects of participation. CASE and CAPE emphasize activity frequency and enjoyment but often lack integration with broader environmental features. These limitations suggest that existing tools may inadvertently narrow the scope of inquiry and obscure the full complexity of the environments shaping occupational participation.

These limitations suggest that existing tools may inadvertently narrow the scope of inquiry and obscure the full complexity of the environments shaping occupational participation. Furthermore, the structure and focus of these instruments may influence which environmental and occupational domains are captured in research. For example, tools such as the PEM-CY and CASE predominantly assess physical and social factors, while offering limited scope to capture structural or systemic dimensions such as economic hardship, discrimination, or policy-driven exclusion. As Villegas et al. [60] argue, interpersonal and institutional discrimination affecting participation among racialized children and families is often not well represented in tools like the PEM-CY, despite being critical to understanding participation disparities. As a result, culturally grounded and economically situated domains may be underreported or oversimplified in the current literature.

Moreover, participation measures often prioritize observable and easily quantifiable domains—such as education and leisure—while overlooking less visible but equally important areas such as health management and rest and sleep. These tool-driven emphases may unintentionally reinforce existing biases about what constitutes “meaningful” participation and where environmental support is most needed. There is a clear need for more inclusive, equity-oriented instruments that can adequately capture the diverse and intersecting barriers experienced by children and families, particularly those from structurally marginalized backgrounds.

The infrequent and selective use of standardized assessment tools has important implications for both research and clinical practice. While valuable for establishing consistency, the dominance of ICF-informed structures may narrow the conceptualization of environment and participation in ways that overlook parental interpretations. A more comprehensive approach is needed—one that recognizes the interdependence of environmental domains and integrates parent perspectives as valid sources of knowledge about participation facilitators and barriers.

Although many studies focused on barriers to participation, several also identified environmental features that supported children’s participation in everyday occupations. Parents highlighted the importance of inclusive school policies, empathetic and flexible educators, community programs that welcomed children with disabilities, and peer relationships that fostered a sense of belonging (e.g., [10,29]). In home and community settings, the presence of accessible infrastructure, family routines that accommodated children’s needs, and collaborative relationships with therapists or service coordinators were also seen as promoting participation. These findings suggest that environmental supports are not only possible but actively operating in many families’ lives, offering a foundation for designing more effective and strength-based interventions.

The findings from this review highlight the importance of supporting occupational participation through a more integrated understanding of the environment. Parents’ narratives point to the need for future research that deliberately addresses all five environmental domains, particularly cultural and economic environments, which were both underrepresented and deeply consequential where present. Studies that only consider physical and social settings offer an incomplete picture of the factors influencing participation. This echoes calls in the literature for more holistic and multidimensional approaches to environmental analysis in childhood disability research [61,62,63].

There is also a need to explore how environmental factors evolve over time and in response to broader societal changes. While a few studies addressed changing environments, such as during the COVID-19 pandemic, most treated environmental influences as static. In the studies that addressed the COVID-19 pandemic, parents reported both barriers and adaptations in participation. Some studies described significant disruptions to education, therapy, and peer interaction, particularly due to school closures, limited online access, or lack of adapted support services [26,34,45]. Some studies highlighted increased family burden and social withdrawal [35,47]. However, a few also noted new opportunities, such as increased home-based participation or flexible scheduling, though these were less frequently emphasized [53]. Comparative studies across different cultural or national contexts, particularly those that examine similar cultural backgrounds in different economic or policy environments, may provide insight into how changeable and context-bound environmental influences truly are.

Crucially, future research should continue to center parental perceptions as a unique and valuable source of evidence. Parents’ insights reveal what enables or constrains participation and how families adapt, resist, or reinterpret their environments for their children’s engagement in everyday occupations. These experiences highlight systemic barriers and unmet support needs that are often overlooked. At the same time, engaging directly with children and adolescents—particularly those approaching adulthood—is also essential. Their first-hand accounts can provide access to participation experiences that occur outside of parental observation, especially in school, peer, and community contexts. Understanding these perspectives requires tools and methods sensitive to cultural, economic, and institutional variation. To advance equitable pediatric healthcare and rehabilitation, future research and policy must prioritize underrepresented environmental domains, integrate standardized assessment tools, and treat caregiver experience as essential for designing inclusive, family-centered services.

This review possesses a number of limitations. Only studies published in English were included, which may have excluded relevant perspectives from non-English-speaking regions. While helpful for mapping the frequency of domain-level attention, the use of vote counting may have oversimplified nuanced findings. Furthermore, while the review focused on studies reporting parental perceptions, the variation in methodological quality and environmental frameworks across the included studies limited the depth of some comparisons.

## 5. Conclusions

This scoping review highlights the importance of environmental factors, including physical, social, institutional, cultural, and economic domains, that parents perceive to shape the occupational participation of children with disabilities. Although physical and social environments are frequently explored, cultural and economic factors are often overlooked, even though, as outlined in the PEO model, they constitute essential dimensions of the environment and were described as influential by parents in several included studies. The findings show that more comprehensive and context-sensitive research is needed, with attention to all environmental domains and a clear emphasis on parents’ perspectives. Improving inclusive and evidence-based practice will require a broader range of research methods, clearer concepts, and a strong commitment to understanding participation as something shaped by complex and changing environmental conditions.

## Figures and Tables

**Figure 1 healthcare-13-01282-f001:**
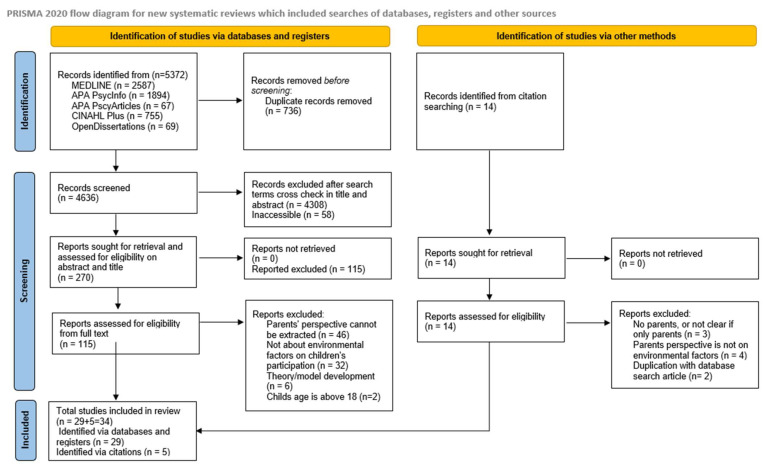
PRISMA flow diagram. Modified from Page et al., BMJ [19].

## Data Availability

The original data presented in the study are openly available in Open Science Framework at https://osf.io/fhpbm/?view_only=7019c3911a5b4588894500575ef55e87 (accessed on 19 April 2025).

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
