# Peer review of "Parental Perspectives on Environmental Factors Affecting Participation of Children with Disabilities: A Scoping Review to Inform Inclusive Healthcare and Support Services"

_healthcare, 2025, doi:10.3390/healthcare13111282_

Round 1
Reviewer 1 Report
Comments and Suggestions for Authors
Thank you for the opportunity to review this scoping review. I believe it will make an important contribution to the literature. Overall, the review was well done and it was a pleasure to read. I will be excited to cite this paper in the future. I have some notes below that may strengthen this submission.
I tend to see the language ‘children with disabilities’ used more often than special needs, but perhaps that is context dependent. I have seen critique of the term special needs, especially from parent advocates, who note that their children’s needs are not exceptional or special – rather they have a right to be included and accommodated in all spaces.
In the population, perhaps you can expand on which children with special needs you are including, for example were there specific diagnosis you included/ excluded, were mental health conditions included (e.g., ADHD), and health conditions (e.g., epilepsy) included?
It sounds as though two reviewers screened each title and abstract. What was the agreement between reviewers? Were there disagreements at the level of full text review that required the discussion with the 3rd reviewer?
In your prisma flow diagram, it seems there were 4636 articles screened after duplicates were removed and 4308 excluded after screening at this level. Given that 4636-4308=328 I am not sure why 270 reports were sought for retrieval?
Can you provide more detail about how textual description, tabulation and vote counting were used in your analysis? I appreciate that additional details about vote counting are provided in Appendix C
There are a couple of formatting things you can resolve in table 2, such as the top row being bolded and the potential to increased the width so that the top row of headings is easier to read.
I am delighted that you considered the potential underreporting of cultural and economic environments given underlying assumptions and ease of observation or measurement. When looking at the PEM-CY we found that interpersonal and institutional discrimination that affects participation for racialized children and families was not well captured in the original measure. I agree that more holistic and inclusive approaches (including measurement tools) are needed to capture that array of environmental influences, especially those that may heighten marginalization (e.g., discriminatory policies, lack of financial security or support, stigma and biases).
Villegas VC, Bosak DL, Salgado Z, Phoenix M, Parde N, Teplicky R, Khetani MA; High Value Early Intervention Research Group. Diversified caregiver input to upgrade the Young Children's Participation and Environment Measure for equitable pediatric re/habilitation practice. J Patient Rep Outcomes. 2023 Aug 28;7(1):87. doi: 10.1186/s41687-023-00627-2. PMID: 37639038; PMCID: PMC10462549.
It was interesting that you used the PEO to organize your study and not the ICF, which also accounts for environmental factors that influence participation. As you noted in your findings most studies that used qualitative designs were informed by the ICF. It may be worth mentioning why you selected the PEO and not the ICF in your introduction.
Suggest rewording “this very little use of assessment methods” to The infrequent use of assessment measures has implications for …
The discussion is well written There were a couple of spots where increased reference to related literature could strengthen your arguments. For example, where you note that future research that addresses all five environmental domains is needed, could you link to others who have made that argument or examples where this has been done well?
Similarly where you note that COVID-19 changed the environmental landscape, were there articles that noted whether that was improved (e.g., increased offering of social and educational programs online that could more easily be accessed by children with disabilities and/or decreased participation due to health risks in social environments).
While I appreciate your focus on parents perspectives, your discussion could also include a call to directly learn from children and youth about their participation experiences, especially as you included children up to age 18, there is a need to learn from youth directly about their participation as it often happens outside of their parents’ purview (e.g., in educational settings or within a peer group).
Author Response
Dear Reviewer 1,
We would like to sincerely thank you for your thoughtful, constructive, and encouraging feedback. We greatly appreciate the time and expertise invested in evaluating our manuscript. The comments provided not only acknowledged the strengths of the work but also offered valuable suggestions that have significantly improved the clarity, structure, and depth of the final version.
In the following document (Please see the attachment), we have addressed each comment point by point. To aid transparency:
-
Blue highlights indicate amendments made in response to Reviewer 1’s comments.
-
Yellow highlights indicate amendments made in response to Reviewer 2’s comments.
-
Purple highlights are used where we responded to comments from both reviewers simultaneously.
-
Khaki highlights reflect updated reference numbers resulting from the revisions.
We hope that the revised manuscript meets the expectations outlined in the review process, and we remain grateful for your insightful and supportive engagement with our work.

Reviewer 2 Report
Comments and Suggestions for Authors
This article is well written and methodologically sound. It's a good contribution, but would be stronger if the discussion had a bit more depth to it. Right now it is more a retelling of what is in the literature, with some synthesis/analysis across the literature.
Page 1 - line 43: The jump to understanding parental perspectives is a big one. It's a valid assertion, but seems disjointed from what has been asserted in the previous statements.
Page 2 - first whole paragraph: More information should be presented early, or at least more clearly, on the environmental factors described by the PEO model. It isn't until late in the paper that the factors are clearly presented.
Just a general comment that there is little that ties or transitions between the ideas presented in the introduction. All points are valid and reasonably presented, but the reader (or at least I) was jumping from one idea to the next and working to tie it together.
Page 4 - line 127: is this study focused solely on parents, or does it include caregivers?
Page 4 - line 172: please clearly provide the inclusion/exclusion criteria. It would also be useful to provide the key words that were used in the database searches earlier.
Page 5 - line 185: would it be reasonable to provide a citation for the JBI Extraction Template?
Page 21 - understanding the results would be facilitated by understanding the inclusion criteria.
Page 21 - line 243: indicates that the participants were parents of children. Should that be parents of children with disabilities? Or were some of these studies focused on parents and not necessarily those of children with disabilities? I would clarify. The inclusion criteria would be useful for understanding this as well.
Page 21 - line 264: Are these five domains all of those in the PEO model? It reads like it is only a subset of 5, but I believe it is all 5. And if it is all 5, make that clear, and then synthesize the nuances.
Page 22 - you provide the number of articles in each of the domains in the previous paragraph, but it would be useful, even if in paratheses, to present the numbers with the narrative for each domain.
Page 24 - I would include some discussion on how the tools used by researchers are contributing to the areas being researched.
Page 26 - line 476: Please provide some information as to how you know that some factors strongly influence participation.
Some of my comments are addressed by the appendices. So you could just provide a reference to the appendices. However, if the appendices are not included with the article in publication, then some information needs to be in the body of the article.
Author Response
Dear Reviewer 2,
We would like to sincerely thank you for your thoughtful, constructive, and encouraging feedback. We greatly appreciate the time and expertise invested in evaluating our manuscript. The comments provided not only acknowledged the strengths of the work but also offered valuable suggestions that have significantly improved the clarity, structure, and depth of the final version.
In the following document (Please see the attachment), we have addressed each comment point by point. To aid transparency:
-
Blue highlights indicate amendments made in response to Reviewer 1’s comments.
-
Yellow highlights indicate amendments made in response to Reviewer 2’s comments.
-
Purple highlights are used where we responded to comments from both reviewers simultaneously.
-
Khaki highlights reflect updated reference numbers resulting from the revisions.
We hope that the revised manuscript meets the expectations outlined in the review process, and we remain grateful for your insightful and supportive engagement with our work.
